The effects of resveratrol feeding and exercise training on the skeletal muscle function and transcriptome of aged rats

Zhou Jing 1 2
Liao Zhiyin 2
Jia Jia 1
Chen Jin-Liang 2
Xiao Qian xiaoqian1956@126.com 2
1 Chongqing Medical and Pharmaceutical College , Chongqing , China
2 Department of Geriatrics, The First Affiliated Hospital of Chongqing Medical University , Chongqing , China
Tatarinova Tatiana
Electronic publication date: 2019 Jul 1
Publication date: 2019
Volume: 7
Electronic Location ID: e7199
Received 2019 Feb 28; Accepted 2019 May 27
Copyright: ©2019 Zhou et al.
Copyright year: 2019
Copyright holder: Zhou et al.
License: This is an open access article distributed under the terms of the Creative Commons Attribution License, which permits unrestricted use, distribution, reproduction and adaptation in any medium and for any purpose provided that it is properly attributed. For attribution, the original author(s), title, publication source (PeerJ) and either DOI or URL of the article must be cited.
License URL: https://creativecommons.org/licenses/by/4.0/

Keywords: Exercise, Resveratrol, Transcriptome, Muscle, Sarcopenia

Funding: Chongqing Natural Science Foundation NO. cstc2018jcyjAX0762 Chongqing Health and Family Planning Commission Foundation NO. 2018MSXM112 Project of Higher Education Teaching Reform of Chongqing NO. 163216 Project of Education Teaching Reform of Chongqing Medical and Pharmaceutical College ygzjg201402 This work was supported by Chongqing Natural Science Foundation (NO. cstc2018jcyjAX0762), Chongqing Health and Family Planning Commission Foundation (NO. 2018MSXM112), Project of Higher Education Teaching Reform of Chongqing (NO. 163216), and Project of Education Teaching Reform of Chongqing Medical and Pharmaceutical College (ygzjg201402). There was no additional external funding received for this study. The funders had no role in study design, data collection and analysis, decision to publish, or preparation of the manuscript.

==============================
This study investigated the effects of resveratrol feeding and exercise training on the skeletal muscle function and transcriptome of aged rats. Male SD rats (25 months old) were divided into the control group (Old), the daily exercise training group (Trained), and the resveratrol feeding group (Resveratrol). After 6 weeks of intervention, the body mass, grip strength, and gastrocnemius muscle mass were determined, and the muscle samples were analyzed by transcriptome sequencing. The differentially expressed genes were analyzed followed by GO enrichment analysis and KEGG analysis. The Old group showed positive increases in body mass, while both the Trained and Resveratrol groups showed negative growth. No significant differences in the gastrocnemius muscle index and absolute grip strength were found among the three groups. However, the relative grip strength was higher in the Trained group than in the Old group. Only 21 differentially expressed genes were identified in the Trained group vs. the Old group, and 12 differentially expressed genes were identified in the Resveratrol group vs. the Old group. The most enriched GO terms in the Trained group vs. the Old group were mainly associated with RNA metabolic processes and transmembrane transporters, and the significantly upregulated KEGG pathways included mucin-type O-glycan biosynthesis, drug metabolism, and pyrimidine metabolism. The most enriched GO terms in the Resveratrol group vs. the Old group were primarily associated with neurotransmitter transport and synaptic vesicle, and the upregulated KEGG pathways included synaptic vesicle cycle, nicotine addiction, retinol metabolism, insulin secretion, retrograde endocannabinoid signaling, and glutamatergic synapse. Neither exercise training nor resveratrol feeding has a notable effect on skeletal muscle function and related gene expression in aged rats. However, both exercise training and resveratrol feeding have strong effects on weight loss, which is beneficial for reducing the exercise loads of the elderly.

Introduction

Sarcopenia is characterized by a reduction in skeletal muscle mass and function during the aging processes (Ryall, Schertzer & Lynch, 2008; Rolland et al., 2008). The pathogenesis of sarcopenia has been proposed to be caused by factors such as oxidative stress, neuromuscular dysfunction, and inflammation (Deschenes, 2004; Kalinkovich & Livshits, 2015; Sousa-Victor & Muñoz Cánoves, 2016; Bano et al., 2017; Zhou et al., 2018). The mechanisms and intervention strategies of sarcopenia, however, have not been thoroughly elucidated to date (Russ et al., 2012; Kalinkovich & Livshits, 2015).

Exercise training is a strategy often adopted in addressing sarcopenia because it may stimulate muscle protein synthesis, activate satellite cells, enhance muscle fiber size, reduce body fat, and improve muscle power (Rolland et al., 2008). However, most of the evidence-based data are from young or middle-aged individuals (Akune et al., 2014). Whether exercise training works well in older populations requires further study. Some previous studies also showed that exercise training can change the expression of several genes (Sakamoto et al., 2005; Thomson & Gordon, 2005). Integrative studies of the effects of exercise training on both muscle function and large-scale gene expression are scarce.

Oxidative stress may be involved in sarcopenia. Whether anti-oxidation can play roles in intervening sarcopenia has not been determined. Resveratrol is a naturally occurring polyphenol that has been suggested to have benefits for humans (Dolinsky & Dyck, 2011). Although the effect of resveratrol on longevity or disease have been debated (Naylor, 2009; Vang et al., 2011), resveratrol has been shown to protect against cellular oxidative stress, improve muscle strength and endurance (Naylor, 2009; Dolinsky et al., 2012; Kaminski et al., 2012; Gordon, Diaz & Kostek, 2013), and increase exercise performance in aged animals (Murase et al., 2009). However, the molecular basis involved with the effects of resveratrol on sarcopenia has been given less attention in the literature. Studies of global gene expression in muscle are needed to evaluate the potential effects of resveratrol on sarcopenia.

Our previous works have observed that the muscle index and the relative grip strength were reduced in aged rats and suggest that the reduced expression of 5′adenosine monophosphate-activated protein kinase (AMPK), insulin-like growth factor 1 (IGF-1), and calcium/calmodulin-dependent serine protein kinase (CASK) can explain the losses of muscle mass and function in aged rats (Liao et al., 2017; Zhou et al., 2018). The objective of the present study was to use RNA-Seq technology to reveal the effects of exercise training and resveratrol feeding on muscle function accompanied by global transcriptomic changes in the muscle of aged rats. The results of this study may contribute to the basal molecular information for sarcopenia interventions.

Materials and Methods

Ethics statement

All animal care and experimental procedures were approved by the Animal Care and Use Committee of Chongqing Medical University (ACUC/CMU/036/2016) and followed the standards for environmental and housing facilities for laboratory animals in China (Gb/T14925-2001). This study uses the method of Zhou et al. (2018), and the method description partly reproduces their wording.

Animals and treatments

Male SD rats (25 months old) were used as experimental animals. Thirty rats were randomly divided into three groups (n = 10 for each group): one group was set as the control (Old), one group was treated with daily exercise training (Trained), and one group was treated with resveratrol feeding (Resveratrol). Animal rearing was processed as previously described in Zhou et al. (2018). Specifically, the rats were maintained in a pathogen-free room with a 12 h/12 h light/dark cycle. The air temperature was controlled at 20 ± 2 °C, and the relative humidity was controlled at 55 ± 5%. All rats were fed ad libitum with standard rat chow, and diets were refreshed once every day. The Old group remained sedentary in their cages for 6 weeks. The Trained group was trained with exercise on a motorized treadmill once at 9:00 daily. The exercise training program was adapted at a speed of 15 cm s−1 for 15 min during the first week, enhanced to a speed of 20 cm s−1 for 20 min during the second week and then enhanced to 30 cm s−1 for 30 min during the next four weeks. During the 6 weeks, the Resveratrol group was fed with oral resveratrol at a dose of 150 mg kg−1d−1 by mixing the resveratrol with the diet. The rats were weighed once weekly, and the ration levels were regulated.

Muscle function measurement

Muscle function was determined as previously described in Zhou et al. (2018). Specifically, at the end of the six-week period, the forelimb grip strength was determined by an electronic grip tester (Cat.47200, Ugo Basil). The strength measurement of each individual was repeated three times, and the maximal value was used as the absolute grip strength (g). The relative grip strength was calculated by dividing the absolute value (g) by the body mass (g). The rats were subsequently transferred back to their cages for three more days of treatment. Then, the rats were euthanized and weighed at 24 h after the last exercise bout/dose of resveratrol. The gastrocnemius muscles of the hindlimbs were quickly sampled and weighed to 0.01 g. The gastrocnemius muscle index (10−3) was calculated by dividing the muscle mass (g) by the body mass (g). Then, the muscle samples from the right hindlimb were frozen in liquid nitrogen and stored in a freezer at −80 °C for transcriptome sequencing.

RNA-Seq library preparation and sequencing analysis

The RNA-Seq library and sequencing analysis were processed as previously described in Zhou et al. (2018). Specifically, muscle samples of three rats from each group were randomly selected for transcriptome sequencing analysis. Total RNA from the muscle tissues was isolated using TRIzol (Invitrogen) and an RNeasy RNA purification kit with DNase treatment (Qiagen). A total amount of 3 g of RNA per sample was used as an input material for the RNA sample preparations. mRNA was purified from the total RNA using poly-T oligo-attached magnetic beads and reverse transcribed into double strand cDNA fragments. Sequencing libraries were generated using the NEBNext® UltraTM RNA Library Prep Kit from Illumina® (NEB, Ipswich, MA, USA) following the manufacturer’s recommendations, and index codes were added to attribute sequences to each sample. The library quality was assessed on the Agilent Bioanalyzer 2100 system. The clustering of the index-coded samples was performed on a cBot Cluster Generation System using the TruSeq PE Cluster Kit v3-cBot-HS (Illumina). The library preparations were sequenced on an Illumina HiSeq platform, and 125-bp/150-bp paired-end reads were generated.

Raw reads in the fastq format were first processed through in-house Perl scripts by removing reads containing adapters, ploy-Ns and low-quality reads from the raw data. Then, Q20, Q30, and GC content were calculated. The reference genome and gene model annotation files were downloaded from the genome website. The reads that passed the quality control were mapped to the Rattus norvegicus genome using Tophat (v2.0.12). HTSeq (v0.6.1) was used to count the read numbers mapped to each gene. Then, the FPKM of each gene was calculated based on the length of the gene, and the read count was mapped to this gene (Trapnell et al., 2010).

Differentially expressed genes and enrichment analysis

The differences in gene expression among groups were analyzed as previously described in Zhou et al. (2018). Specifically, differential expression analysis was performed using the DESeq R package (1.18.0). DESeq provided statistical routines for determining the difference in the digital gene expression data using a model based on the negative binomial distribution. The resulting P-values were adjusted using Benjamini and Hochberg’s approach for controlling the false discovery rate. Gene Ontology (GO) enrichment analysis of differentially expressed genes was implemented by the GOseq R package, in which the gene length bias was corrected. GO terms with a corrected P-value less than 0.05 were considered to be significantly enriched by differentially expressed genes. KOBAS software was used to test the statistical enrichment of the differentially expressed genes in the KEGG pathways.

Quantitative real-time PCR

Quantitative real-time PCR was performed as previously described in Zhou et al. (2018). Specifically, 1.5 g of extracted total RNA was added to a final volume of 20 l to synthesize cDNA using a PrimeScriptTM RT Reagent Kit with gDNA Eraser (Takara Biomedical Technology, Beijing, China). Eight muscular function-related genes (Prkaa-2, FOXO3, IGF-1, CASK, TRIM67, IL-6, MGMT, SCN5a) were selected, and the coding mRNA quantity was analyzed by the ABI VIIATM7 Real-Time PCR system (Applied Biosystems, Foster City, CA) with a quantitative real-time PCR kit (GoTaq® qPCR Master Mix, Promega Biotech, Beijing, China). Amplification was performed for 1 cycle at 95 °C for 10 min, followed by 40 cycles at 95 °C for 15 s, and 60 °C for 30 s with 1 pM primers (Table 1). qRT-PCR for all genes was performed in triplicate, with glyceraldehyde-3-phosphate dehydrogenase (GAPDH) serving as the reference gene. Relative expression levels were calculated with the 2−ΔΔCT method and were presented as fold changes compared to the Old group.

Table 1 Primers used in the qPCR.

Gene	Foward Primer (5′–3′)	Reverse Premer (3′–5′)	
FOXO3	AACAGTACCGTGTTCGGACC	AGTGTCTGGTTGCCGTAGTG	
CASK	CGAGCTTCCTGAGCACCGAG	CAGCGTGGAGGGTTCTGAAA	
IGF-1	CGTACCAAAATGAGCGCACC	CCTTGGTCCACACACGAACT	
IL-6	GCAAGAGACTTCCAGCCAGT	GTCTCCTCTCCGGACTTGTG	
MGMT	AAAGCCGTCTGGCTGAAATTG	CCAGGACACATGCAGCTCTTTTAAT	
PrKaa-2	GATCGGACACTACGTGCTGG	ACTGCCACTTTATGGCCTGT	
SCN5a	CTGACTATAGCCGCAGCGAA	GGCTCATCTGCAGAGCTAGG	
TRIM67	TCGTCCAGATGAAATTGCCAC	CCAGGGTGGCGCTGTTATTA	
GAPDH	AACTCCCATTCCTCCACCTT	GAGGGCCTCTCTCTTGCTCT	

Statistical analysis

Parameters of the body and muscle were calculated using Microsoft Excel 2003 (Microsoft Corporation, Redmond, WA, USA) and analyzed using SPSS 11.5 (SPSS Inc., Chicago, IL, USA). Initial body mass, body mass gain, and specific growth rate among the groups were compared using one-way ANOVA followed by Duncan’s test. Final body masses and absolute grip strengths among the groups were also compared with initial body mass as a covariate. A paired-samples t-test was used to compare the initial mass with the final body of each group. Differences were considered significant at the level of P < 0.05. Data are presented as the mean ± s.e.m.

Results

Body mass and muscle function

The final body masses of the Trained group and Resveratrol group were significantly lower than those of the Old group with initial body mass as a covariate (P < 0.001) (Table 2). The final body masses of the Old group were 1.94% larger but not significantly different from their initial body masses (t = 0.921, P = 0.388). Both the Trained group and the Resveratrol group showed decreases in body mass. The final body masses of the Trained group were 12.2% lower than their initial body masses (t =  − 8.763, P < 0.001), while the final body masses of the Resveratrol group were 5.31% lower than their initial body masses (t =  − 2.543, P = 0.039). Body mass gain and specific growth rate were significantly larger in the Old group compared to both the Trained group and Resveratrol group. The gastrocnemius muscle indexes for the left limb and the right limb were 4.81 × 10−3 and 4.88 × 10−3 for the Old group, 5.10 × 10−3 and 5.38 × 10−3 for the Trained group, and 4.90 × 10−3 and 4.97 × 10−3 for the Resveratrol group (Fig. 1). The gastrocnemius muscle index was not significantly different among the three groups (left limb: F = 0.399, P = 0.676; right limb: F = 2.984, P = 0.072). The absolute grip strengths were 1,206, 1,329, and 1,297 g for the Old, Trained, and Resveratrol rats, respectively, and were not significantly different among the three groups (F = 1.203, P = 0.320) (Fig. 2A). However, the analysis of covariates showed significant differences among the three groups, with final body mass as a covariate (F = 6.488, P = 0.008). Similarly, the relative grip strengths were significantly higher in both the Trained group and Resveratrol group than in the Old group (F = 5.019, P = 0.017) and were 2.04, 2.62, and 2.33 g g−1 for the Old, Trained, and Resveratrol rats, respectively (Fig. 2B).

Table 2 Body mass parameters of the three groups of aged rats.

Parameters	Old	Trained	Resveratrol	F	P	
Initial body mass (g)	586.0 ± 33.5	591.1 ± 30.3	591.0 ± 24.9	0.010	0.990	
Final Body mass (g)	596.6 ± 34.7	518.4 ± 34.7	559.5 ± 26.9	14.558	<0.0001	
Body mass gain (g)	10.6 ± 11.5a	−72.8 ± 8.3c	−31.5 ± 12.4b	14.663	<0.0001	
Specific growth rate (% day−1)	0.04 ± 0.04a	−0.31 ± 0.03c	−0.13 ± 0.05b	16.613	<0.0001	
Notes.

Data were expressed as mean ± s.e.m.

a,b,c Different superscripts in each row indicate significant differences among groups (P < 0.05).

Old control aged rats

Trained the aged rats treated with six weeks of exercise training

Resveratrol the aged rats treated with six weeks of resveratrol feeding

Figure 1 Gastrocnemius muscle indexes of the hindlimbs of the rats.

Data are presented as the mean ± s.e.m. Blue: left hindlimb; red: right limb.Old: control aged rats; Trained: the aged rats treated with six weeks of exercise training; Resveratrol: the aged rats treated with six weeks of resveratrol feeding.

Figure 2 Comparisons of the absolute grip strength and relative grip strength between the control rats and the rats treated with exercise training and resveratrol feeding.

Data are presented as the mean ± s.e.m. *: significantly different from the control rats (P < 0.05). (A) the absolute grip strength; (B) the relative grip strength. Old: control aged rats; Trained: the aged rats treated with six weeks of exercise training; Resveratrol: the aged rats treated with six weeks of resveratrol feeding.

Transcriptome sequencing and assembly

Three cDNA libraries were built from the skeletal muscle of each rat treatment. The parameters of the reads are shown in Table S1, and the reads have been uploaded to the NCBI Sequence Read Archive (accession number: SRA605175). The expression level correlated strongly among treatments (Fig. 3).

Figure 3 Correlations among the expression level of the muscle tissue samples of the control rats and the rats treated with exercise training and resveratrol feeding.

a, b, c: The three replicate samples of each group. Old: control aged rats; Trained: the aged rats treated with six weeks of exercise training; Resveratrol: the aged rats treated with six weeks of resveratrol feeding.

Trained vs. Old: Only 21 differentially expressed genes were identified between the Trained group and the Old group (Table 3, Fig. S1), within which 14 genes (e.g., ribosomal RNA processing seven homolog A (Rrp7a), RT1 class I, locus CE9, pseudogene 1 (RT1-CE9-ps1), the rRNA promoter binding protein (LOC257642), protein YIF1B (LOC103689986), and potassium sodium-activated channel subfamily T member 1 (Kcnt1)) were downregulated, while seven genes (e.g., slingshot protein phosphatase 2 (SSH2), polypeptide N-acetylgalactosaminyltransferase 5 (GALNT5), RGD1562339 (RGD1562339), EFR3 homolog B (EFR3b), zinc finger protein 474 (ZFP474), mitochondrial carrier triple repeat 1 (MCART1), and cytidine deaminase-like (LOC100909857)) were upregulated in the Trained group compared to those in the Old group. Seventy GO enrichment terms were enriched in the Trained group vs. the Old group (Table S2). The twenty most enriched GO terms were primarily associated with RNA metabolic processes and transmembrane transporters (Table 4). KEGG analysis identified several significantly upregulated pathways in the Trained group vs. Old group, including mucin-type O-Glycan biosynthesis, drug metabolism, and pyrimidine metabolism, but with only one differentially expressed gene (Table S4). The eight selected genes (Prkaa-2, FOXO3, IGF-1, TRIM67, IL-6, MGMT, and SCN5a) were verified by quantitative real-time PCR, within which only two genes (CASK and TRIM67) showed significant differences in the Trained group vs. the Old group (Fig. 4).

Table 3 Several differentially expressed genes in gastrocnemius muscle tissue of the three groups of aged rats.

Gene symbol	Gene full name	Log2 Fold change	Corrected-P Value	
Trained vs Old			
Down-regulated				
RRP7a	ribosomal RNA processing 7 homolog A	−2.623	0.000	
RT1CE9ps1	RT1 class I, locus CE9, pseudogene 1	−2.453	0.050	
LOC257642	rRNA promoter binding protein	−2.005	0.050	
LOC103689986	protein YIF1B	−1.902	0.021	
KCNT1	potassium sodium-activated channel subfamily T member 1	−1.878	0.030	
RRP8	ribosomal RNA processing 8, methyltransferase, homolog (yeast)	−1.563	0.005	
DBP	D-box binding PAR bZIP transcription factor	−1.509	0.002	
PALLDL1	palladin-like 1	−1.469	0.050	
Up-regulated				
SSH2	slingshot protein phosphatase 2	1.330	0.021	
GALNT5	polypeptide N-acetylgalactosaminyltransferase 5	1.373	0.042	
RGD1562339	RGD1562339	1.647	0.050	
EFR3b	EFR3 homolog B	1.764	0.005	
ZFPp474	zinc finger protein 474	2.076	0.006	
MCART1	mitochondrial carrier triple repeat 1	2.549	0.019	
LOC100909857	cytidine deaminase-like	3.260	0.005	
Resveratrol vs Old			
Down-regulated				
ZMYND19	zinc finger, MYND-type containing 19	−1.262	0.000	
Up-regulated				
ALDH1a1	aldehyde dehydrogenase 1 family, member A1	1.213	0.000	
TRIM67	tripartite motif-containing 67	3.198	0.000	
SYT1	synaptotagmin 1	3.726	0.000	
LOC100910446	syntaxin-7-like	4.571	0.000	
SNAP25	synaptosomal-associated protein 25	5.168	0.000	
TTR	transthyretin	5.377	0.000	
SLC17a7	solute carrier family 17 member 7	6.599	0.000	
NELLl2	neural EGFL like 2	Inf	0.000	
Notes.

Old control aged rats

Trained the aged rats treated with six weeks of exercise training

Resveratrol the aged rats treated with six weeks of resveratrol feeding

Table 4 The most enriched Gene Ontology terms of the three groups of aged rats.

GO accession	Description	Term type	P Value	Up	Down	
Trained vs. Old					
GO:0034660	ncRNA metabolic process	biological_process	0.002697	0	3	
GO:0006364	rRNA processing	biological_process	0.005946	0	2	
GO:0016072	rRNA metabolic process	biological_process	0.00631	0	2	
GO:0042254	ribosome biogenesis	biological_process	0.0165	0	2	
GO:0034470	ncRNA processing	biological_process	0.017902	0	2	
GO:0022613	ribonucleoprotein complex biogenesis	biological_process	0.035357	0	2	
GO:0000139	Golgi membrane	cellular_component	0.026633	1	1	
GO:0022857	transmembrane transporter activity	molecular_function	0.04874	0	3	
Resveratrol vs. Old					
GO:0006836	neurotransmitter transport	biological_process	7.01E–05	3	0	
GO:0001505	regulation of neurotransmitter levels	biological_process	7.68E–05	3	0	
GO:0042572	retinol metabolic process	biological_process	8.95E–05	2	0	
GO:0034308	primary alcohol metabolic process	biological_process	0.000159	2	0	
GO:0048278	vesicle docking	biological_process	0.000258	2	0	
GO:0016050	vesicle organization	biological_process	0.000291	3	0	
GO:0001523	retinoid metabolic process	biological_process	0.000298	2	0	
GO:0006066	alcohol metabolic process	biological_process	0.000356	3	0	
GO:0016101	diterpenoid metabolic process	biological_process	0.000399	2	0	
GO:0022406	membrane docking	biological_process	0.000415	2	0	
GO:0006721	terpenoid metabolic process	biological_process	0.000528	2	0	
GO:0031201	SNARE complex	cellular_component	2.35E–06	3	0	
GO:0008021	synaptic vesicle	cellular_component	3.31E–05	3	0	
GO:0060076	excitatory synapse	cellular_component	8.74E–05	2	0	
GO:0098793	presynapse	cellular_component	0.000105	3	0	
GO:0030672	synaptic vesicle membrane	cellular_component	0.00026	2	0	
GO:0042734	presynaptic membrane	cellular_component	0.000312	2	0	
GO:0045202	synapse	cellular_component	0.000328	3	1	
GO:0000149	SNARE binding	molecular_function	2.37E–05	3	0	
GO:0017075	syntaxin-1 binding	molecular_function	4.45E–05	2	0	
Notes.

Old control aged rats

Trained the aged rats treated with six weeks of exercise training

Resveratrol the aged rats treated with six weeks of resveratrol feeding

Figure 4 Quantitative real-time PCR verifications for the selected genes of the control rats and the rats treated with exercise training and resveratrol feeding.

Data are presented as the mean ± s.e.m. *: significantly different from the control rats (P < 0.05). Old: control aged rats; Trained: the aged rats treated with six weeks of exercise training; Resveratrol: the aged rats treated with six weeks of resveratrol feeding.

Resveratrol feeding vs. Old: Only 12 differentially expressed genes were identified between the Resveratrol group and the Old group (Table 3, Fig. S2), within which two genes were downregulated and 10 genes (e.g., aldehyde dehydrogenase 1 family, member A1 (ALDH1a1), tripartite motif-containing 67 (TRIM67), synaptotagmin 1 (SYT1), syntaxin-7-like (LOC100910446), synaptosomal-associated protein 25 (SNAP25), transthyretin (TTR), solute carrier family 17 member 7 (SLC17a7), neural EGFL-like 2 (NELL2), apolipoprotein C-I-like (LOC100910181) were upregulated in the Resveratrol group compared to those in the Old group. We found 219 GO terms that were enriched in the Resveratrol group vs. the Old group (Table S3). The most enriched GO terms were mainly associated with neurotransmitter transport and synaptic vesicle (Table 4). KEGG analysis identified several significantly upregulated pathways in the Resveratrol group vs. the Old group, including synaptic vesicle cycle, nicotine addiction, retinol metabolism, insulin secretion, retrograde endocannabinoid signaling, and glutamatergic synapse, but with only one differentially expressed gene (Table S4). Only the synaptic vesicle cycle pathway was significantly enriched with more than one differentially expressed gene (Table 5). Only CASK and TRIM67 showed significant differences between the Resveratrol group and the Old group by quantitative real-time PCR verification (Fig. 4).

Discussion

Effects of exercise

Even though exercise was suggested as a potential strategy in addressing age-related sarcopenia, previous studies obtained conflicting results on the effect of exercise on muscle mass and function. Many studies have found that exercise can ameliorate the aging-related loss of muscle mass and function and change muscle function-related gene expression (Sakamoto et al., 2005; Thomson & Gordon, 2005; Rolland et al., 2008; Shefer et al., 2010; Cisterna et al., 2016). However, the present study showed no positive effects of exercise on muscle mass and absolute strength (Figs. 1 and 2), which may result from heterogeneity in subject characteristics and the exercise type and intensity (Pattanakuhar et al., 2017). A moderate intensity and duration was used in the exercise training program in our study. More studies are needed to test whether exercise works better with stronger intensity or longer duration. In addition, our results showed that the relative grip strength of the Trained group increased compared to the Old group, which could be primarily due to the remarkable effect of exercise training on weight loss.

Table 5 The significant KEGG enriched pathways the rats treated with six weeks of resveratrol feeding compared to the control rats.

Pathway	KEGG ID	Input number	Background number	Corrected P-Value	
Resveratrol vs Old up-regulated				
Synaptic vesicle cycle	rno04721	3	63	0.000	
Notes.

Old control aged rats

Resveratrol the aged rats treated with six weeks of resveratrol feeding

One explanation for the loss of muscle mass and function in aging rats is the reduced activities of AMPK, a fundamental regulator of energy metabolism (Salminen, Hyttinen & Kaarniranta, 2011; Hardie & Ashford, 2014; Hardman et al., 2014). It has been reported that exercise may lead to AMPK activation (Sakamoto et al., 2005). However, the exercise-induced phosphorylation status of AMPK was observed only in the soleus muscle but not in the plantaris muscle of aged rats (Thomson & Gordon, 2005). Some studies found no significant exercise-induced upregulation of AMPK expression in the muscle of aged rats (Reznick et al., 2007). The present study showed no positive effects of exercise on the expression of muscle function-related genes, e.g., AMPK, FOX3, and IGF-1 (Figs. 1, 2 and 4; Table 3). In addition, the expression levels of CASK, sodium and calcium channel regulation-related genes (Eichel et al., 2016; Nafzger & Rougier, 2017) were even reduced in the aged rats after exercise training.

Recently, sixteen genes have been identified as grip strength determining genes in humans (Willems et al., 2017); however, none of these genes was regulated in aged rats by exercise training in the present study. In fact, only a limited number of genes were upregulated in the Trained group compared to the Old group, and several transport- or transcript-related genes were even downregulated in the Trained group, e.g., RRP7a, RT1CE9ps1, and KCNT1. Our results suggest that exercise training has limited effects on the expression of genes in the skeletal muscle of aged rats. However, our results presented only data on the transcriptomic response. More studies on other levels, such as protein expression, metabolites and hormone dynamics, are necessary to test the effects of exercise on aging-related sarcopenia.

Effect of resveratrol

Similar to the effects of exercise training, dietary resveratrol has remarkable effects on weight loss in aged rats. Consistently, it has been reported that resveratrol (300 and 750 mg/kg/d) reduced the body weight of rats aged 6 weeks during the 13-week feeding period (Williams et al., 2009). In addition, resveratrol (400 mg/kg/d) reduced the body weight of mice aged 8 weeks during the 9-week feeding period, which was attributed to the increase in energy expenditure (Lagouge et al., 2006). However, resveratrol at a dose of 400 mg/kg/d did not significantly affect the body weight of rats during the 2-week feeding period (Momken et al., 2011), and resveratrol at a dose of 60 mg/kg/d did not affect body weight in mice aged 28 months during the 10-month feeding period (Jackson, Ryan & Alway, 2011). That the body weight was reduced by a relatively higher dose and longer period (Lagouge et al., 2006 and the present results) but not by a lower dose or shorter period (Jackson, Ryan & Alway, 2011; Momken et al., 2011) suggests that the effects of dietary resveratrol on body mass depend on both its dose and duration of administration. Consistently, it has been suggested that the effects of resveratrol on in vitro muscle cell plasticity are dose-dependent: low resveratrol doses promoted in vitro muscle regeneration and attenuated the impact of ROS, while high doses blocked the regenerative process (Bosutti & Degens, 2015).

Resveratrol has been shown to alter fatty acid and glucose metabolism, inhibit protein degradation, and protect against cellular oxidative stress (Naylor, 2009; Kaminski et al., 2012; Gordon, Diaz & Kostek, 2013). In the present study, dietary resveratrol did not change the muscle mass or absolute strength of the aged rats, and the apparent increase in the relative strength should be attributed to the weight loss effects of resveratrol in the aged rats. The effects of dietary resveratrol on skeletal muscle mass and function remain diverse in previous studies. Resveratrol reduced the aging-related declines in muscle mass and function of mice at 18 weeks of age (Murase et al., 2009; Dolinsky et al., 2012) but not of mice at 28 months of age (Jackson, Ryan & Alway, 2011). Additionally, dietary resveratrol attenuated the decrease in muscle mass and strength caused by mechanical unloading in young rats (∼250 g) (Momken et al., 2011) but was unable to attenuate the reductions in muscle mass in aged rats (32 months) (Bennett, Mohamed & Alway, 2013). This finding implies that resveratrol may affect the skeletal muscle differently in the old individuals compared to the young ones and may have limited effects in attenuating sarcopenia.

Resveratrol activates a number of signaling pathways, including the expression of IGF-1, AKT, AMPK, and SIRT1, in myoblasts (Barger et al., 2008; Murase et al., 2009; Naylor, 2009; Park et al., 2012; Montesano et al., 2013; Ligt, Timmers & Schrauwen, 2015). However, our study showed no effects of dietary resveratrol on the expression of muscular function-related genes. Similarly, no effects of resveratrol on gene expression profiles were observed in normal mice, and it was explained that collecting tissue samples from the objectives fasted overnight may mask the effects of resveratrol on its target gene expression (Yoshino et al., 2012), which could be a reason for the unchanged expression of those genes in rats by resveratrol in our study, since the rats were fasted overnight before tissue sampling.

Interestingly, however, resveratrol induced the expression of several genes, including ALDH1a1, TRIM67, SYT1, SNAP25, TTR, SLC17a7, NELL2, LOC100910446, and LOC100910181 (Table 3). Among these genes, most of them (e.g., SYT1, LOC100910446, SNAP25, SLC17a7) are related to neuron synaptosome function (Tucker & Chapman, 2002; Yoshihara & Montana, 2004; Santos, Li & Voglmaier, 2009; Antonucci et al., 2016). NELL2, a gene involved in neural cell growth and differentiation, was also upregulated in aged mice fed with resveratrol. The enhanced expression of these genes suggests an upregulation of synaptic vesicle cycle signaling in aged mice fed with resveratrol. SLC17a7 encodes vesicular glutamate transporter 1 (VGLUT1), which is associated with the functions in transporting the excitatory neurotransmitter glutamate into synaptic vesicles and has important roles in the neuromuscular synapse in nervous systems and neuromuscular junctions of muscles (Varoqui et al., 2002; Kraus, Neuhuber & Raab, 2004). Even though the functional significance of VGLUT1 has not been determined, VGLUT1 expression has been found to be altered by the aging process and several diseases (Jung et al., 2018). A recent study found that retinal VGLUT1 protein and gene expression were decreased in diabetic mice, indicating changes in the signal transmission from photoreceptors to bipolar cells and among postreceptoral neurons (Ly et al., 2014). This study also found that the decreased expression of VGLUT1 could not be normalized by metformin treatment (Ly et al., 2014). The positive effects of resveratrol on SLC17a7 expression imply a potential application of diabetic retinopathy intervention.

An expression of ed inat rosci creased expression of 1. Another finding of the present study was that a tripartite motif family gene (TRIM67) was upregulated in aged mice fed resveratrol. The TRIM protein family has been implicated in many biological processes, including cell differentiation, apoptosis, transcriptional regulation and signaling pathways, and plays an important role in the broader immune response (McNab et al., 2011), especially in the restriction of infection by lentiviruses (Ozato et al., 2008). Overexpression of TRIM proteins can increase the membrane repair capacity of muscular dystrophy and restore muscle function and morphology (Alloush & Weisleder, 2013; Dahl-Halvarsson et al., 2018). TRIM67 is highly expressed in the brain and regulates neuritogenesis, brain development, and behavior (Yaguchi et al., 2012; Boyer et al., 2018). Our results suggest that resveratrol has positive effects on the expression of TRIM67 in the muscle of aged rats, implying that the underlying mechanisms warrant further study.

Conclusions

In conclusion, the results present the global transcriptomic information involved in exercise and anti-oxidation interventions of skeletal muscle function in aged rats. Neither exercise training nor resveratrol feeding has a remarkable effect on skeletal muscle function and related gene expression in aged rats. Interventions with other exercise programs and resveratrol doses are needed, and studies at the protein level are also necessary to test whether exercise and resveratrol interventions are effective in addressing aging-related sarcopenia. However, both exercise training and resveratrol feeding have remarkable effects on weight loss, which is beneficial for suppressing exercise loads in the elderly. Our findings should be verified by future studies in human subjects.

Supplemental Information

Table S1 Summary of the raw sequencing data of the gastrocnemius muscle tissue of the rats treated with six weeks of exercise training and resveratrol feeding compared to the control rats

Old: old rat; Trained: old rat treated by exercise training; Resveratrol: old rat treated by oral resveratrol; a, b, c: Three replicate samples of each treatment.

Click here for additional data file.

Table S2 The significant Gene Ontology terms of the rats treated with six weeks of exercise training compared to the control rats

Old: old rat; Trained: old rat treated by exercise training; GO: Gene Ontology

Click here for additional data file.

Table S3 The significant Gene Ontology terms of the rats treated with six weeks of resveratrol feeding compared to the control rats

Old: old rat; Resveratrol: old rat treated by oral resveratrol; GO: Gene Ontology.

Click here for additional data file.

Table S4 The significant KEGG enriched pathways the rats treated with six weeks of exercise training and resveratrol feeding compared to the control rats

Old: old rat; Trained: old rat treated by exercise training; Resveratrol: old rat treated by oral resveratrol.

Click here for additional data file.

Figure S1 Volcanoplot of differential expressed genes between exercise training compared to the control rats

Old: old rat; Trained: old rat treated by six weeks of exercise training.

Click here for additional data file.

Figure S2 Volcanoplot of differential expressed genes between resveratrol feeding compared to the control rats

Old: old rat; Resveratrol: old rat treated by six weeks of oral resveratrol.

Click here for additional data file.

Data S1 Raw data of the three groups of rats

Click here for additional data file.

We thank Ms. Yuxing Zhao and Die Pu for their help in animal care. We sincerely thank the anonymous reviewers for their helpful suggestions and comments.

Additional Information and Declarations

Competing Interests

Author Contributions

Animal Ethics

DNA Deposition

Data Availability

The authors declare there are no competing interests.

Jing Zhou conceived and designed the experiments, performed the experiments, analyzed the data, contributed reagents/materials/analysis tools, prepared figures and/or tables, approved the final draft.

Zhiyin Liao and Jin-Liang Chen performed the experiments, contributed reagents/materials/analysis tools.

Jia Jia performed the experiments, analyzed the data, prepared figures and/or tables, authored or reviewed drafts of the paper.

Qian Xiao conceived and designed the experiments, performed the experiments, analyzed the data.

The following information was supplied relating to ethical approvals (i.e., approving body and any reference numbers):

The experiment was approved by the Animal Care and Use Committee of Chongqing Medical University (ACUC/CMU/036/2016).

The following information was supplied regarding the deposition of DNA sequences:

The sequences are accessible at NCBI SRA: SRA605175.

The following information was supplied regarding data availability:

Data for this study are available in Figshare: Zhou, Jing (2019): Data of transcriptome, realtime PCR, and strength of muscle of old rats. figshare. Dataset. https://doi.org/10.6084/m9.figshare.7781066.v2.

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
