# Peer review of "The effects of resveratrol feeding and exercise training on the skeletal muscle function and transcriptome of aged rats"

_PeerJ, doi:10.7717/peerj.7199_

## Round 0.1 · original submission · Major Revisions

It is a very interesting manuscript, and the presented results can be useful for development of sarcopenia treatment. Although, all animal studies must be taken with caution, since not all animal models are applicable to humans. If you could describe limitations of your study in the discussion, it would help the readers a lot. I also suggest expanding legends of tables and figures.

Possibly, a professional technical writer or a native English speaking scientist would be able to help to avoid sentences like this: "Thirty male SD rats aged 25 months (old)", "Differential expression genes were analyzed", etc.

Some of the readers tend to skip materials and methods part. For their sake it would be better not to start the results with abbreviations ("The body masses of the OE rats and OR rats were significantly lower than those of the OC rats"). Reviewer #2 suggested not using abbreviations at all, since too many of them are confusing for the reader.

It would benefit the paper if you could add additional results, such as: protein classical exercise markers in all groups, time from exercise to muscle harvesting, lean to fat mass changes, concentration of hormones (Reviewer #2).

Reviewer #1 suggested several recent studies that support validity of your results. Reviewer #1 also made an interesting observation, that while the treatment of resveratrol did not change the overall muscle function, it did show that there was an overall decrease in body mass index compared to untreated sedentary control animals. These data suggest that while the dose given may not be useful for muscle function, it may be useful as a treatment for maintaining weight or for weight loss.

Reviewer 1 ·

Basic reporting

See general comments

Experimental design

See general comments

Validity of the findings

See general comments

Additional comments

The manuscript entitled ‘The impacts of resveratrol feeding and exercise training on skeletal muscle transcriptome and function of old rats’ by Zhou et al., has investigated the role of resveratrol as a potential treatment for sarcopenia.

The introduction is concise and clearly outlines the relevance of this study.

The methods are well thought out and extensively described in the materials and methods. The authors carefully describe the qRT-PCR experiments carried out (Line 148 on), however the authors do not include the primers used for these experiments and these should be included.

In the first paragraph of the results the authors state that there is a change in body mass of each experimental group, however no data is actually presented. It is suggested that the data be elaborated upon in the main text (i.e. such as percentage changes etc). Additionally, Table S1 is important data and should be included as a main table rather than supplement.

Moreover, more details of the actual data for the grip strength tests should be included in the results section to demonstrate there is or is not a difference. While this is presented in the figure, this should also be addressed in the main body of text.

In the discussion the authors introduce the term AMPK, however this should be defined and why it is important.

There has been a recent study (Bosutti and Degans, Sci. Reps. 2015, 5) which has shown that there is a dose-dependent effect of resveratrol on muscle cell plasticity. This is an important study and the dose of resveratrol given may alter the overall outcome of this study. While it is not expected that the work be repeated to take this into account, it is important to mention this in the discussion.

The authors seem to have missed an important point in the data presented. While the treatment of resveratrol did not change the overall muscle function, it did show that there was an overall decrease in body mass index compared to untreated sedentary control animals. These data suggest that while the dose given may not be useful for muscle function, it may be useful as a treatment for maintaining weight or for weight loss. This is consistent with previous studies where there is potential protection from high fat diets (Momken, et al., FASEB J, 2011, 25, 10) but does not improve metabolic function in non-obese individuals (Yoshino et al., Cell Metabolism, 2012, 16, 658). Interestingly, a large number of the genes found to be altered are normally thought of as neuron specific (with some lower expression in other tissues in the EST database), but these are linked to diabetes and obesity. For example: Slc17a7 (Ly et al., Diabetologica, 2014, 57, 192), TRIM67 (Comuzzie et al., PLos One, 2012, 7, e51954).

It is suggested the authors include a section in the discussion to address this and the potential for further study.


Specific comments:

Overall the language is fine throughout the manuscript but there are places where it is difficult to interpret the precise meaning (highlighted in annotated manuscript).

Line 105: The term executed should be replaced with euthanized.

Line 221-223 – there is a change in font compared the rest of the manuscript.

Annotated reviews are not available for download in order to protect the identity of reviewers who chose to remain anonymous.

Reviewer 2 ·

Basic reporting

no comment

Experimental design

no comment

Validity of the findings

no comment

Additional comments

Zhou et al. sought to characterise the functional and transcriptomic response to resveratrol and exercise on old rats. Identifying effective treatments for the aging process is an important area of research, with sarcopenia contributing to aging-related mortality. Old rat studies are logistically difficult, and the authors should be commended for their efforts in this study. There are some limitations that should be addressed before the manuscript is published.
Major
It is hard to conclude from the transcriptomic data whether the exercise or resveratrol interventions were effective. It is not unsurprising that the authors did not observe differences in the classical markers of exercise via transcriptomics as the majority of the transcriptomic response to exercise is acute. The authors should assess the protein expression of some of the classical exercise markers. Since several of the pathways are somewhat similar in response to resveratrol, the authors should also assess these markers in the resveratrol treated group.
Please specify how long after the last exercise bout/dose of resveratrol the muscles of the rats were harvested.
Several of the enriched GO and KEGG pathways/terms appear to have very few altered genes. A cut-off should be used to exclude pathways with only 1 differential gene.
The most striking effect of the interventions appears to be weight-loss. It would benefit the manuscript if the authors were able to provide further characterisation of this weight-loss. E.g. lean-mass or fat-mass changes. Also beneficial would be plasma concentration of hormones associated with adipose e.g. adiponectin, leptin, insulin; or appetitive eg. GLP-1.
Minor
Conclusions in the discussion regarding the effects of exercise in older rats are overstating the data given the limitations of the study. Please temper these assertions.
The authors appear to treat ‘exercise’ as a homogenous intervention. However, it is extremely varied with many modalities and intensities, please take this into account when discussing the predominantly negative data in the current study. I.e. it would be pertinent to discuss more intense or longer duration forms of exercise interventions.
It is not necessary to abbreviate the names of the intervention groups and does not aid the reader. A suggestion of the names would be ‘Old’, ‘Trained’, and ‘Resveratrol’.
Please proof-read the manuscript carefully for grammatical and spelling errors. Some examples of grammatical mistakes below:
• Line 57: This sentence is not grammatically correct.
• Line 60: ‘Diseases’ should not be pluralised
• Line 65: this sentence is not grammatically correct.
• Line 233: ‘literatures’ should not be pluralised.

---

## Round 0.2 · accepted · Accept

The paper presents interesting results about effects of exercise and resveratrol feeding. Both treatments resulted in weight loss in aged rates. However, the study showed no positive effects of exercise on muscle mass and gene expression. The authors presented their results in clear and objective manner, making a strong case for the need of additional investigation in this area.

Reviewer 1 ·

Basic reporting

See below

Experimental design

See below

Validity of the findings

See below

Additional comments

The manuscript language has greatly improved over the last submission. It is now in standard English and easy to follow.

The authors have addressed all the comments from the review and incorporated the changes into their manuscript.

The authors have expanded their discussion to include more references that are relevant to the study. They have also re-written the discussion to be broader and more relevant to their work. This study does contain some interesting results that warrant further study in the future (i.e. the role of neuronal markers in muscle function). The authors have also stated other important aspects that need to be taken into account in addition to genomic approaches - i.e. gene expression, hormone response etc.

Overall, this version is a significant improvement and provides interesting data that will contribute to the field.

Reviewer 2 ·

Basic reporting

n/a

Experimental design

n/a

Validity of the findings

n/a

Additional comments

I am satisfied with the amendments and recommend this manuscript be published.